# OpenReview forum: "Transferable and Stealthy Adversarial Attacks on Large Vision-Language Models"
_ICLR.cc/2026/Conference — ICLR 2026 Poster_

### Official Review · Reviewer_aN48 · 2025-10-19

**Soundness:** 3
**Presentation:** 3
**Contribution:** 3
**Rating:** 6
**Confidence:** 3

**Summary:**

This paper proposes PSI, an  adversarial attack method targeting large Vision-Language Models (VLMs) that achieves strong transferability, naturalness, and stealthiness. To enhance transferability and naturalness, PSI leverages diffusion priors to better align adversarial examples with the natural image distribution, and adopts a progressive alignment strategy to alleviate overfitting to a single fixed surrogate objective. To further improve stealthiness, PSI incorporates source-aware cues during the denoising process, thereby preserving visual fidelity and avoiding easily detectable artifacts. Experimental results demonstrate that PSI produces adversarial examples with superior imperceptibility to humans and reduced detectability by VLMs, outperforming existing attack methods.

**Strengths:**

1. The progressive alignment objectives in PSI, constructed through co-evolving selection on localized regions, appear to be a novel and interesting contribution.
2. The incorporation of source-aware denoising is also an interesting aspect of PSI, as it helps preserve the visual consistency of the original image while enhancing stealthiness.
3. The experimental results demonstrate promising performance across various VLMs.

**Weaknesses:**

1. Although the experiments demonstrate that PSI achieves strong transferability, the concern raised in Eq. (3) remains only partially addressed, as the proposed joint objective is still constructed based on the surrogate model  𝐹.
2. The contribution of the source-aware denoising component is not explicitly evaluated or ablated in the experimental section, making it difficult to assess its individual impact.
3. The defense experiments are not comprehensive. The authors are encouraged to consider additional defense techniques, such as DISCO [1], to provide a more complete evaluation of PSI’s robustness.
4. Minor Comment: In line 261, DDIM should be corrected to DDPM.

Reference

 [1] C.-H. Ho and N. Vasconcelos, “DISCO: Adversarial defense with local implicit functions,” in Proc. Adv. Neural Inf. Process. Syst., vol. 35, Jan. 2022, pp. 23818–23837.

**Questions:**

1. The authors are encouraged to include a theoretical analysis to support the effectiveness of PSI, instead of relying exclusively on experimental validation.
2. The authors are advised to report and compare the computational cost of PSI with other baseline methods, to better illustrate its efficiency and practicality.

---

> ### Author Response · Authors · 2025-11-20
> **Response to Reviewer aN48 (Part 1)**
>
> We sincerely appreciate the reviewer's valuable comments. We hope that our response can address the raised questions. Please let us know if there are any further questions.
>
> >**W1: Joint objective still constructed based on surrogate model $\mathbf{F}$.**
>
> Our threat model assumes a strict black-box victim LVLM, where the attacker has no access to parameters, gradients, architecture, training data, or even API queries. Under this setting, it is fundamentally infeasible to construct a training objective directly based on the victim model.
> Therefore, using a surrogate model $\mathbf{F}$ becomes the only practically accessible source of differentiable guidance and approximate alignment behavior.
>
> On top of this, our joint objective additionally incorporates the naturalness term $p_{\mathcal D}\big(x\big)$ to improve transferability, rather than merely depending on the alignment term $p_{\mathbf{F}}\big(f^{\text{tar}} \mid x^{\text{adv}}\big)$ which alone risks overfitting to $\mathbf{F}$.
>
>
> >**W2: Request for ablation on source-aware denoising component.**
>
> Source-aware denoising primarily contributes to the stealthiness of the generated images. As shown in the following table, the non-reference BRISQUE scores remain comparable across settings, whereas the reference-based LPIPS metric shows a substantially larger improvement when source-aware denoising is applied.
>
> | Method                            | ASR (GPT-5)  | S-ASR (GPT-5) | BRISQUE | LPIPS |
> |-----------------------------------|------|-------|---------|-------|
> | PSI                               | 78.6 | **62.8**  | **22.14**   | **0.192** |
> | PSI *w/o* source-aware denoising  | **81.0** | 57.0  | 23.6    | 0.241 |
>
>
> >**W3: Request for more comprehensive experiments on defenses.**
>
> We evaluate PSI’s robustness against existing defenses from three perspectives:
>
> i) Adversarial training.
> As shown in Table 1, adversarially trained models such as FARE$^{4}$ remain ineffective against PSI. This is likely because PSI introduces perturbations with a different magnitude and structure compared to conventional pixel-level attacks used during training.
>
> ii) Purification-based defenses.
> As illustrated in Table 2, purification methods can reduce the success rate of all baseline attacks, while PSI is more robust to purification methods. However, these defenses incur additional computational overhead and suppress fine-grained perceptual details.
>
> iii) LVLM-based detection.
> Model-based detection mechanisms can also identify adversarial examples produced by all methods, including PSI as shown in Table 4 in Appendix B.4, but they require extra queries and complex analysis pipelines.
>
> DISCO [1] is categorized as a purification-based defense. We will update our comments after finishing the corresponding experiments.
>
> >**W4: Word 'DDIM' in Line 261 is incorrect.**
>
> We respectfully clarify that the DDIM reference in Line 261 is correct. Methods such as AdvDiffVLM use DDIM inversion to encode the source image into a single latent state. In contrast, PSI is designed in the DDPM noise space and embeds the source information not only in the latent but also in the noise injected at each denoising step, enabling source-aware denoising.

---

> ### Author Response · Authors · 2025-11-20
> **Response to Reviewer aN48 (Part 2)**
>
> >**Q1: Request for theoretical analysis on PSI.**
>
> We thank the reviewer for raising the question of theoretical analysis. While most components of PSI are motivated and validated empirically, we now provide a simple theoretical argument explaining why PSI is designed to distribute perturbations uniformly along the denoising trajectory, rather than injecting them at a single latent step as in ACA [2].
>
> **Proposition.** Spreading small perturbations across multiple timesteps yields a better joint objective than concentrating the entire perturbation at a single step.
>
> *Proof Sketch.* A reverse-diffusion sampler runs for $T$ steps. At step $t$ we inject a small perturbation $\delta_t$. Let $a_t \ge 0$ be the effectiveness of step $t$ on the final output. The first-order alignment gain is:
> $$
> \Delta \text{Align} \approx \sum_{t=1}^T a_t |\delta_t|.
> $$
>
> Since the diffusion score is close to zero near the model’s denoised predictions, the first-order term of the naturalness measure becomes negligible, and the deviation is therefore dominated by its second-order component. Accordingly, we approximate the naturalness variation as:
> $$
> \Delta \text{Nat} \approx \sum_{t=1}^T w_t |\delta_t|^2,
> $$
> where $w_t \ge 0$.
>
> Maximizing the joint objective amounts to minimizing $\Delta \text{Nat}$ for the same total effective contribution $\Delta \text{Align}$. Let  $ S := \Delta \text{Align} $ be fixed. By Cauchy–Schwarz inequality,
> $$
> \Big(\sum_{t=1}^T a_t |\delta_t|\Big)^2
> = \Big(\sum_{t=1}^T \frac{a_t}{\sqrt{w_t}} \cdot \sqrt{w_t} |\delta_t|\Big)^2
> \le
> \Big(\sum_{t=1}^T \frac{a_t^2}{w_t}\Big)
> \Big(\sum_{t=1}^T w_t |\delta_t|^2\Big).
> $$
>
> Therefore,
> $$
> \sum_{t=1}^T w_t |\delta_t|^2
> \ge
> \frac{S^2}{\sum_{t=1}^T \tfrac{a_t^2}{w_t}},
> $$
> with equality if and only if $|\delta_t| \propto \frac{a_t}{w_t}$.
>
> In contrast, concentrating all perturbation on a single step $\tau$ corresponds to the extreme allocation with $|\delta_\tau| = S/a_\tau$ and $|\delta_t| = 0$ for $t \neq \tau$, which strictly increases $\sum_t w_t |\delta_t|^2$ whenever $T>1$. Hence, under the joint objective, spreading small perturbations across multiple timesteps yields better transferability than injecting them at a single step.
>
>
>
> >**Q2: Request for computational cost of PSI.**
>
> Under the same number of iterations as set in our experimental configuration, PSI does not incur significant additional computational overhead compared to baseline methods, e.g., AttackVLM. In contrast, FOA requires substantial extra time due to the clustering computation. The following table reports the per-image generation time of adversarial examples on a single NVIDIA A800 GPU.
>
> | Method          | AttackVLM | AdvDiffVLM | M-Attack | FOA | PSI |
> |-----------------|-----------|------------|----------|-----|-----|
> | Time per sample (sec) | 52        | 63         | 56       | 113 | 70  |
>
>
>
> **Reference**
>
> [1] C.-H. Ho and N. Vasconcelos, “DISCO: Adversarial defense with local implicit functions,” in Proc. Adv. Neural Inf. Process. Syst., vol. 35, Jan. 2022, pp. 23818–23837.
>
> [2] Chen, Z., Li, B., Wu, S., Jiang, K., Ding, S., & Zhang, W. (2023). Content-based Unrestricted Adversarial Attack. NeurIPS 2023.

---

> > ### Comment · Reviewer_aN48 · 2025-11-25
> >
> > The authors’ responses have addressed most of my concerns. Based on the overall assessment as well as the comments from other reviewers, I will maintain my current ratings.

---

### Official Review · Reviewer_EmtG · 2025-10-26

**Soundness:** 2
**Presentation:** 3
**Contribution:** 3
**Rating:** 6
**Confidence:** 4

**Summary:**

This paper proposes **Progressive Semantic Infusion (PSI)**, a diffusion-based adversarial attack designed to generate adversarial examples that are both **highly transferable** and **visually imperceptible**. By combining diffusion priors, progressive semantic alignment, and source-aware denoising, PSI ensures both naturalness and strong attack capability. Experiments conducted on multiple open-source and commercial VLMs demonstrate that PSI achieves superior performance compared to FOA, M-Attack, and AdvDiffVLM in terms of transferability and stealthiness.

**Strengths:**

* **Important and realistic problem.**
 The paper addresses one of the most critical and practical challenges in current VLM security — achieving black-box adversarial robustness with both transferability and visual imperceptibility.
* **Well-motivated and coherent design.**
 The method leverages diffusion priors to maintain naturalness and progressive alignment to enhance cross-model consistency, forming a conceptually clear and technically sound framework.
* **Comprehensive experiments.**
 The paper includes extensive experiments across multiple datasets and models, with strong baselines, ablation studies, and hyperparameter analyses that substantiate the effectiveness of PSI.

**Weaknesses:**

* **Unclear boundary of contributions.** The method can be seen as a combination of diffusion-based purification and FOA-like alignment. The novelty is somewhat incremental, and the paper should include a dedicated section clarifying the key theoretical and methodological contributions beyond prior works such as FOA and AdvDiffVLM.
* **Limited validation of the naturalness term.** The joint objective includes a data distribution term \( p
_D(x) \), which is only indirectly approximated via the diffusion prior. The paper lacks analysis or evidence showing how this term contributes to improved transferability or robustness.
* **Unjustified choice of scale parameter.** The subregion scale \( s \in [0.4, 0.9] \) is used without ablation or justification. In contrast, FOA and M-Attack adopt \( s \in [0.5, 0.9] \). The rationale for this difference should be explained, ideally supported by quantitative sensitivity analysis.
* **Insufficient theoretical support for the “degenerate case” statement.** The claim that *“The random cropping techniques used in M-Attack and FOA can be viewed as a degenerated case of the progressive alignment”* is conceptually intuitive but lacks formal theoretical justification or mathematical derivation.

**Questions:**

1. **Clarify the main contributions.** Add a short subsection explicitly summarizing PSI’s conceptual and theoretical advances over FOA, AdvDiffVLM, and M-Attack.
2. **Provide empirical or theoretical evidence for the naturalness term.** Analyze how the naturalness loss \( \mathcal{L}_{nat} \) correlates with attack transferability or perceptual quality, supported by visual or statistical comparisons.
3. **Include a hyperparameter ablation on the scale range.** Report results under different \( s \) intervals (e.g., [0.2, 0.8], [0.5, 0.9]) to justify the chosen values and their influence on ASR and perceptual metrics (LPIPS, BRISQUE).
4. **Strengthen theoretical grounding.** Provide a formal explanation or appendix derivation showing how random cropping in M-Attack and FOA can be mathematically reduced to a special case of progressive alignment (e.g., when temporal dynamics or semantic references are removed).
5. **Enhance clarity and reproducibility.** Include detailed pseudocode, hyperparameter tables, and mask-update procedures for local alignment in the appendix or supplementary materials to facilitate reproducibility.

---

> ### Author Response · Authors · 2025-11-20
> **Response to Reviewer EmtG**
>
> We sincerely appreciate the reviewer's valuable comments. We hope that our response can address the raised questions. Please let us know if there are any further questions.
>
> >**Q1 & W1: Request to reclarify contributions and differences with previous works.**
>
> Differences from prior work arise in formulation, inversion usage, and adversarial optimization.
>
> i) PSI incorporates a joint objective that models alignment and naturalness for better transferability. In contrast, most existing methods focus solely on alignment.
>
> ii) PSI employs source-aware denoising by leveraging an edit-friendly noise space to ensure stealthiness. In comparison, AdvDiffVLM use DDIM inversion combined with label-dependent GradCAM masking.
>
> iii) PSI injects adversarial semantics through progressive alignment with co-evolving region selection, achieving more stable and effective alignment. FOA and M-Attack use random cropping that ignores the content layout of the source and target.
>
> These differences collectively enable PSI to deliver stronger stealthiness and transferability, as demonstrated in our experiments.
>
>
> >**Q2 & W2: Request for validation on the naturalness term.**
>
> Traditional methods only emphasize the alignment $p_{\mathbf{F}}\bigl(f^{\text{tar}} \mid x^{\text{adv}}\bigr)$ on the surrogate VLMs. Our joint objective additionally incorporates naturalness term $p_\mathcal{D}\bigl(x\bigl)$ to pursue better transferability. We would like to explain our motivation in two aspects:
>
> i) Since both surrogate and target LVLMs are trained on large-scale natural-image datasets, the natural image distribution serves as their shared training distribution. Consequently, adversarial examples that remain within this distribution are more likely to preserve alignment on the victim model than out-of-distribution (OOD) ones.
>
> ii) We validate this motivation using PRO [1], an OOD detector based on robustness to perturbations, to examine the influence of the naturalness term. We group all baseline methods mentioned in Section 5.1 according to their PRO scores, where lower scores indicate lower naturalness. The following table shows a clear positive correlation between PRO and average attack success rate (on GPT-5).
>
> | Naturalness Bin     | PRO         | ASR   |
> |---------------------|-------------|-------|
> | Low                 | [0, 0.3)    | 14.7% |
> | Medium              | [0.3, 0.6)  | 36.2% |
> | High                | [0.6, 1]    | 52.3% |
>
> We have added this analysis in Appendix B.5 of the revised version.
>
> >**Q3 & W3: Request for ablation study on scale range.**
>
> Our scale range differs from FOA and M-Attack because our region-selection mechanism is different from theirs. FOA and M-Attack use random cropping and therefore ignore the content layout of the source and target images. In contrast, PSI employs co-evolving region selection, which provides more semantically consistent and fine-grained source–target matching at each step. Consequently, the optimal scale range for PSI differs from that of FOA and M-Attack.
>
> Moreover, PSI is not sensitive to the choice of scale range as shown in the following table.
>
> | Scale Range | ASR (GPT-5) | ASR (Grok-4) | BRISQUE | LPIPS  |
> |-------------|------------:|-------------:|--------:|-------:|
> | $[0.2, 0.9]$  |        75.2 |         79.9 |   23.15 |  0.212 |
> | $[0.4, 0.9]$  |     **78.6**|     **81.4** |   22.14 |**0.192**|
> | $[0.6, 0.9]$  |        76.4 |         77.6 | **21.31**|  0.199 |
>
>
>
> >**Q4: Why random-cropping is a degenerated case of PSI?**
>
> Thanks for the question. We elaborate on this below:
>
> i) Random cropping pairs a randomly selected target region with a randomly selected source region at each step, ignoring the inherent content layout present in both images.
>
> ii) Progressive Alignment accounts for the content layout of both the target and source images via co-evolving selection. It prioritizes semantically dominant regions on the target and gradually expands to the full image as optimization proceeds. Meanwhile, it selects source regions with higher feature similarity to stabilize the alignment process.
>
> Therefore, progressive alignment without co-evolving selection reduces to a random cropping strategy.
>
> >**Q5: Request for pseudocode, hyperparameter tables and mask-update procedures.**
>
> To enhance clarity, we update the detailed pseudocode of our Progressive Semantic Infusion (PSI) in Appendix A.2 of the revised version. Hyparameters of PSI are reported in Section 5.1 of the main paper. We respectfully clarify that PSI does not adopt any mask-update mechanism used in AdvDiffVLM. Instead, we ensure stealthiness through a source-aware denoising process, as illustrated in the pseudocode.
>
> **Reference**
>
> [1] Chen, Wenxi, et al. "Leveraging Perturbation Robustness to Enhance Out-of-Distribution Detection." Proceedings of the Computer Vision and Pattern Recognition Conference(CVPR). 2025.

---

### Official Review · Reviewer_orSX · 2025-10-31

**Soundness:** 1
**Presentation:** 1
**Contribution:** 1
**Rating:** 2
**Confidence:** 5

**Summary:**

This paper aims to improve the effectiveness and imperceptibility of adversarial examples against large vision–language models (VLMs) by leveraging diffusion models with progressive alignment. Experiments demonstrate that the proposed method outperforms existing approaches.

**Strengths:**

1. Enhancing the stealthiness of adversarial attacks is an important and timely research direction.
2. The idea of aligning the distribution between adversarial and natural examples is reasonable and intuitive.
3. Experiments on multiple models validate the effectiveness of the proposed approach.

**Weaknesses:**

1. The paper appears to focus on targeted attacks, but this is not clearly stated by the authors. This lack of clarity may mislead readers regarding the nature of the adversarial setting.
2. The literature review is limited. Many existing adversarial attack methods are relevant, beyond those specifically developed for VLMs. Their core ideas could also apply to this context. The authors should discuss these works and include comparative results.
3. The motivation behind the Joint Objective (e.g., Eq. (4)) is not well explained. The paper should clarify its intuition and necessity.
4.  The proposed method only considers attacks on the image modality while ignoring the text modality. Since a key characteristic of VLMs is their cross-modal alignment, it is not meaningful to attack or evaluate the model solely through the image modality.
5. The authors propose to select the source model that is similar to the target model, which is not reasonable.

**Questions:**

Please refer to Weaknesses.

---

> ### Author Response · Authors · 2025-11-20
> **Response to Reviewer orSX**
>
> We appreciate the reviewers' feedback and hope our responses can address the raised concerns. Please let us know if there are any further questions.
>
> >**W1: The adversarial setting (targeted vs. untargeted) was considered unclear.**
>
> In the Problem Statement (Line 130), we state that our attack is targeted toward a given target image. In other words, our attack aims to induce the LVLM to produce responses regarding the target image. This setting is widely used in works including AnyAttack [1], M-Attack [2] and FOA [3]. For distinction, some other attacks (e.g., CroPA [4]) aim to force the model to output a specific phrase such as 'unknown'.
>
>
> >**W2: The literature review was considered limited.**
>
> This work is closely related to Vision-Language Models (VLMs) and transfer-based adversarial attacks in VLMs, which have been reviewed in the main text. Due to page limitations, we provide additional review of unrestricted adversarial attacks in Appendix C. We further expand the review of classical adversarial attacks in Appendix C of the revised version.
> It's worth noting that non-VLM attacks follow different threat models and evaluation protocols, making a direct comparison less meaningful in our setting.
>
>
> >**W3: Motivation behind the joint objective is not well explained.**
>
> Traditional methods only emphasize the alignment $p_{\mathbf{F}}\bigl(f^{\text{tar}} \mid x^{\text{adv}}\bigr)$ on the surrogate VLMs. Our joint objective additionally incorporates naturalness term $p_\mathcal{D}\bigl(x\bigl)$ to improve transferability. We would like to explain our motivation in two aspects:
>
> i) Since both surrogate and target LVLMs are trained on large-scale natural-image datasets, the natural image distribution serves as their shared training distribution. Consequently, adversarial examples that remain within this distribution are more likely to preserve alignment on the victim model than out-of-distribution (OOD) ones.
>
> ii) We additionally validate this motivation using PRO [5], an OOD detector based on robustness to perturbations, to examine the influence of the naturalness term to attack success rate (ASR) on the victim black-box model. We group all baseline methods mentioned in Section 5.1 according to their PRO scores, where lower scores indicate lower naturalness. The following table shows a clear positive correlation between PRO and average attack success rate (on GPT-5), indicating better transferability.
>
> | Naturalness Bin     | PRO         | ASR   |
> |---------------------|-------------|-------|
> | Low                 | [0, 0.3)    | 14.7% |
> | Medium              | [0.3, 0.6)  | 36.2% |
> | High                | [0.6, 1]    | 52.3% |
>
> We have added this analysis in Appendix B.5 of the revised version.
>
>
> >**W4: Attacks on the input text should also be considered.**
>
> Thanks. Restricting the adversary to the image modality means that the attacker only needs access to the image and does not require permission to modify the user’s text prompt. This assumption is widely used and represents a realistic setting where the attacker can only manipulate images but cannot alter user-provided prompts. We thus focus on attacks in the visual modality, following the commonly adopted setting in recent works like AnyAttack, M-Attack, and FOA.
>
>
> >**W5: Unreasonable selection of surrogate models.**
>
> We respectfully disagree with the reviewer and like to clarify the rationality of our selection:
>
> i) The selection of surrogate models is consistent with the settings adopted in recent works like M-Attack and FOA. We thus follow this commonly adopted setup to ensure the fair comparison.
>
> ii) Although both surrogates and victim models are based on Transformers, they differ in architectural details, model scale, and accessibility. We demonstrate that open-source models can successfully serve as surrogates to attack black-box commercial systems, including GPT-5 and Grok-4. This setup reflects a realistic threat model in practice.
>
> **Reference**
>
> [1] Jiaming Zhang, et al. "Anyattack: Towards large-scale self-supervised adversarial attacks on vision-language models." Proceedings of the Computer Vision and Pattern Recognition Conference(CVPR), pp. 19900–19909, June 2025b.
>
> [2] Li, Zhaoyi, et al. “A Frustratingly Simple yet Highly Effective Attack Baseline: Over 90% Success Rate Against the Strong Black-Box Models of GPT-4.5/4o/o1.” arXiv preprint arXiv:2503.10635, 2025.
>
> [3] Jia, Xiaojun, et al. “Adversarial Attacks Against Closed-Source MLLMs via Feature Optimal Alignment.” arXiv preprint arXiv:2505.21494, 2025.
>
> [4] Luo, Haochen, et al. "An image is worth 1000 lies: Adversarial transferability across prompts on vision-language models." Proceedings of the International Conference on Learning Representations (ICLR), 2024.
>
> [5] Chen, Wenxi, et al. "Leveraging Perturbation Robustness to Enhance Out-of-Distribution Detection." Proceedings of the Computer Vision and Pattern Recognition Conference(CVPR). 2025.

---

> > ### Comment · Reviewer_orSX · 2025-11-27
> > **Response to authors**
> >
> > Thanks for the authors' efforts in addressing my concerns. However, several issues remain. First, I do not agree that the introduction of a new model type justifies disregarding prior adversarial attack methods. The paper does not sufficiently investigate or relate to the existing literature. Second, the choice of surrogate models plays a crucial role in attack effectiveness, yet the authors do not clarify the key differences between the source and target models, making it difficult to assess the claimed adversarial transferability. For these reasons, I will keep my original score.

---

> > > ### Author Response · Authors · 2025-11-28
> > > **Follow-up Response to Reviewer orSX**
> > >
> > > Thank you for the follow-up. We respond to the remaining issues as follows. Please let us know if there are any further questions.
> > >
> > > > **Issue 1:** The paper disregards prior adversarial methods.
> > >
> > > i) We respectfully clarify that the main text focuses on the most relevant areas, namely Large Vision-Language Models (VLMs) and transfer-based adversarial attacks on VLMs due to the limited space. We include the review of classical adversarial attacks and unrestricted attacks in Appendix C. According the reviewer's suggestion, we have added a review of adversarial attack on Vision-language pre-trained (VLP) models with new references including ETU [1] and Co-Attack [2]:
> > >
> > > [1] Zhang, Peng-Fei, Zi Huang, and Guangdong Bai. "Universal adversarial perturbations for vision-language pre-trained models." Proceedings of the 47th International ACM SIGIR Conference on Research and Development in Information Retrieval. 2024.
> > >
> > > [2] Zhang, Jiaming, Qi Yi, and Jitao Sang. "Towards adversarial attack on vision-language pre-training models." Proceedings of the 30th ACM International Conference on Multimedia. 2022.
> > >
> > > In VLP benchmarks, both images and captions come from the dataset rather than user prompts. In contrast, LVLMs rely on user-provided prompts that attackers cannot alter, making image-only perturbations the realistic and widely adopted threat model.
> > >
> > > ii) Classical adversarial attacks are designed for closed-set classification, VLP attacks operate on feature-space alignment for retrieval or matching, whereas LVLMs produce open-set, free-form textual outputs. These three settings follow different threat models and evaluation protocols. Therefore, non-VLM attacks cannot be directly applied or fairly compared in the VLM attack scenario. Consistent with this distinction, recent works on VLM attack (e.g., AnyAttack, M-Attack, FOA, etc.) likewise do not use non-VLM attacks as baselines.
> > >
> > >
> > > > **Issue 2:** The key differences between the surrogate and the target models is unclear.
> > >
> > > Our surrogate models include three variants of CLIP, following the setup used in M-Attack and FOA. The differences are as follows:
> > >
> > > i) Different accessibility: All surrogate models are publicly available. In contrast, victim models include black-box commercial systems, such as GPT-5 and Grok-4, which are not open-source.
> > >
> > > ii) Different architecture: our surrogate models are standalone CLIP dual encoders,
> > > whereas the victim models are end-to-end LVLMs such as MiniGPT-4 and Grok-4. For open-source LVLMs, their vision towers differ from our chosen CLIP surrogate models. For commercial LVLMs, the vision stack is proprietary and unknown to us, so our attacks do not rely on any assumed overlap in the vision encoder.
> > >
> > > Taken together, these differences show that we can attack black-box
> > > commercial LVLMs using only open-source, low-cost surrogate models,
> > > without any access to victim models' internal architecture or training data. We will include those discussions in the main text to clarify this concern.

---

### Official Review · Reviewer_CND6 · 2025-10-31

**Soundness:** 3
**Presentation:** 3
**Contribution:** 3
**Rating:** 6
**Confidence:** 4

**Summary:**

- This paper proposes PSI, a transferable adversarial attack against large VLMs.
   - PSI leverages diffusion models to progressively align and infuse natural target semantics during the denoising process.
- To enhance transferability, diffusion priors are utilized to align adversarial examples more closely with the natural image distribution, while progressive alignment mitigates overfitting to a single surrogate objective.
- In addition, PSI introduces source-aware cues via DDPM inversion to preserve visual fidelity and avoid perceptible artifacts.
- Experimental results demonstrate that PSI effectively attacks open-source, adversarially trained, and commercial VLMs, including GPT-5 and Grok-4, achieving superior performance in both transferability and stealthiness compared to existing methods.

**Strengths:**

- The paper introduces a transferable adversarial attack tailored for large Vision-Language Models (VLMs), revealing a critical semantic-level vulnerability in modern multimodal systems and providing valuable insights toward building more robust and trustworthy models.
- The proposed PSI method is conceptually simple yet effective, achieving superior attack performance across various open-source and commercial models. It consistently outperforms multiple state-of-the-art approaches in both transferability and stealthiness.
- The paper offers a clear discussion of related work and experimental results, with comprehensive implementation details and prompts provided in the appendix, thereby strengthening reproducibility and transparency.

**Weaknesses:**

- The source-aware denoising module primarily builds upon the existing DDPM inversion technique [1], which is not a novel contribution of this work and therefore should not be highlighted as a core innovation.

> [1] An Edit-Friendly DDPM Noise Space: Inversion and Manipulations, CVPR 2023.

- There is some inconsistency in the experimental results. For example, the reported ASR on Claude-3.5 differs considerably from that in M-Attack, even though both studies use the same perturbation budget $\epsilon=16/255$ and GPTScore threshold (0.3). The paper would benefit from a clearer explanation of these discrepancies to ensure comparability and reproducibility.

- The main methodological difference from prior approaches such as M-Attack, FOA, and AdvDiffVLM lies in the design of a distinct localized alignment strategy combined with a co-evolving selection mechanism within the diffusion framework. While these modifications lead to performance gains, they appear incremental in nature, and the paper provides limited analysis or ablation evidence to justify the effectiveness of each proposed component.

**Questions:**

- It would be beneficial for the paper to include a clearly structured Threat Model section that explicitly defines the attack assumptions, adversary capabilities, and experimental settings. This addition would help readers better understand the scope and applicability of the proposed method.

- The evaluation of defenses could be further strengthened. Presenting results on a broader range of Large Vision-Language Models (LVLMs) would provide stronger evidence of the generality and robustness of the proposed PSI attack.

- Could the authors clarify what specific modifications or innovations were introduced beyond the original inversion framework? For example, was any new modeling applied to noise injection or update dynamics during inversion?

> An Edit-Friendly DDPM Noise Space: Inversion and Manipulations, CVPR 2023.

- The paper highlights progressive alignment and co-evolving selection as the core innovations of PSI. While an ablation study is provided (Table 3), it is limited to a single model, which may not be sufficient to fully demonstrate the general effectiveness of these components. Could the authors consider expanding the ablation analysis or providing additional quantitative evidence to better validate their impact on transferability and stealthiness?

- The reported ASR on Claude-3.5 is notably higher than in M-Attack, despite using the same perturbation budget $\epsilon=16/255$ and GPTScore threshold (0.3). Were there any differences in prompts, sampling steps, or surrogate model ensembles? How sensitive are your results to these settings?

- PSI leverages diffusion priors to enhance naturalness, but stronger adherence to the natural image manifold might reduce controllability over the target semantics. Did the authors observe such a trade-off, and if so, is there a mechanism in PSI to balance naturalness and semantic fidelity?

- DiffPure or semantic-consistency detection) perform against PSI, and what potential directions exist for developing defense strategies tailored to such diffusion-based attacks?

---

> ### Author Response · Authors · 2025-11-20
> **Response to Reviewer CND6 (Part 1)**
>
> We sincerely appreciate the reviewer's valuable comments. We hope that our response can address the raised questions. Please let us know if there are any further questions.
>
> >**Q1: Need for a well-defined threat model.**
>
> We thank the reviewer for suggesting a clear threat model. We adopt the threat model commonly used in existing works like M-Attack, FOA, and AdvDiffVLM. Below are the detailed descriptions of the threat model:
>
> Attacker's Goal:
> The attacker aims to craft an adversarial image that causes the victim LVLM produce outputs similar to those generated from the target image. At the same time, the adversarial image should avoid revealing its adversarial nature at both the input and output levels, as illustrated in Figure 1(e) of the main paper.
>
> Attacker’s Knowledge:
> The attacker has no access to the victim LVLM's parameters, gradients, architecture, training data, or API queries. The attacker has access to an open-source surrogate vision–language model (e.g., CLIP).
>
> Attacker's Capability:
> The attacker can only craft and distribute malicious images that will later be consumed by the victim LVLM in downstream applications. The attacker cannot modify any text prompts, system instructions, or other non-visual inputs.
>
> We have included a Threat Model section in Appendix A.1 of the revised version to ensure the self-consistency of our work.
>
> >**Q2: Request for experiments on more LVLMs.**
>
> As requested by the reviewer, we conducted additional experiments on open-sourced, adversarially robust and commercial LVLMs, as shown in the table below. In the table, each cell reports the corresponding attack’s attack success rate (ASR) and stealthy attack success rate (S-ASR). It is clear that, our method shows promising performance.
>
>
> | **Attacks**    | LLaVA-1.5-7B     | Qwen2.5-VL-7B      | TeCoA$^4$ [1] (Adv. Robust) | GPT-4o          |
> |----------------|:----------------:|:-------------------:|:------------------------:|:----------------:|
> | AttackVLM      |  9.9 /  9.6      |   8.5 /  8.1        |       0.9 /  0.8         |    3.1 /  2.8    |
> | AdvDiffVLM     | 33.5 / 33.2      |  27.8 / 26.4        |      14.2 / 13.9         |   13.4 /  9.2    |
> | AnyAttack      | 37.0 / 28.1      |  30.2 / 23.6        |      12.7 / 10.2         |   25.1 / 11.5    |
> | M-Attack       | 85.5 / 81.2      |  82.4 / 79.2        |      55.6 / 52.0         |   75.7 / 56.2    |
> | FOA            | **87.8** / 82.6  |  86.0 / 80.5        |      58.3 / 55.3         |   77.7 / 58.2    |
> | **PSI (ours)** | 87.2 / **84.4**  |  **86.5** / **83.0**        |   **67.7 / 66.1**        | **80.6 / 64.7**  |
>
>
>
> >**Q3 & W1: Request to clarify the novelty of PSI's inversion process.**
>
> Thanks, we agree that inversion technique is not new. Our contribution lies in the novel way we use inversion to construct transferable and stealthy adversarial examples.
>
> i) We leverage the diffusion prior from diffusion-based inversion to preserve naturalness in the joint objective, while existing works always overlook diffusion prior.
>
> ii) Our source-aware denoising is conducted in an edit-friendly DDPM noise space, so that every timestep update carries source-image cues, enabling full-trajectory perturbation. In contrast, prior work either relies on DDIM inversion with label-dependent masks, or techniques such as null-text inversion, which cannot support PSI's full-trajectory perturbation.
>
>
> >**Q4 & W3: Request for more analysis and ablations to support progressive alignment.**
>
> We provide the following extended ablation table on progressive alignment, where each cell reports the attack’s ASR and S-ASR:
>
> | Ablation Setting              | GPT-5 | Gemini-2.5  | Grok-4  | Claude-3.5  | BRISQUE |
> |-------------------------------|----------------------|----------------------------|------------------------|----------------------------|----------|
> | PSI (ours)                | 78.6 / 62.8          | 75.8 / 71.5                | 81.4 / 75.0            | 21.8 / 15.2                | 22.14    |
> | *w/o* progressive alignment | 22.8 / 15.0          | 18.9 / 14.4                | 24.5 / 20.5            | 6.6 / 4.3                  | 22.28    |
> | *w/o* co-evolving selection | 71.3 / 52.5          | 69.6 / 65.5                | 72.8 / 57.3            | 19.6 / 13.2                | 25.60    |
>
> As shown, without progressive alignment, using a single fixed alignment objective significantly reduces transferability. Moreover, co-evolving selection provides more stable source–target correspondences, thereby further improving attack success rates.
> According to the suggestion, we have added this result to Appendix B.6 of the revised version.

---

> ### Author Response · Authors · 2025-11-20
> **Response to Reviewer CND6 (Part 2)**
>
> >**Q5 & W2: How our experimental setting differs from M-Attack, and the sensitivity of this setting.**
>
> Our experimental setting differs from M-Attack in both data scale and evaluation protocol:
>
> i) M-Attack reports results on only 100 images, whereas we evaluate on the full NIPS 2017 challenge dataset, yielding a more stable and reliable result.
>
> ii) Text prompts used in M-Attack are not unified: GPT uses a 20-word limit while Gemini uses a 25-word limit.
> In contrast, we standardize all evaluations using the same prompt,
> “Describe this image in 30 words.”
> A relaxed word limit could better simulate real-world scenarios and expose output-level adversarial cues, as demonstrated in Figure 6 in the Appendix. This, in turn, enables a fair and consistent assessment of stealthiness.
>
> iii) The following table reports ASR / S-ASR for each model, and the results show mild sensitivity to the choice of prompt. Therefore, we emphasize that evaluations across different models and attacks should use a unified prompt to avoid misleading comparisons.
>
> | Model (Attack)       | 15 words    | 30 words    |
> |----------------------|:-----------:|:-----------:|
> | GPT‑5 (M‑Attack)     | 80.9 / 69.7 | 73.8 / 54.5 |
> | Grok‑4 (M‑Attack)    | 85.3 / 78.2 | 77.9 / 70.5 |
> | Claude-3.5 (M-Attack)| 16.6 / 14.2 | 12.4 /  9.8 |
>
> Claude is substantially harder to attack, which aligns with findings in M-Attack and FOA.
>
>
> >**Q6: What is the trade-off between naturalness and target semantic control, and how does PSI balance it?**
>
> i) Thanks for the insightful comment. We do observe a tradeoff between naturalness and controllability over the target semantics. We measure the controllability over the target sematics through the global similarity between the adversarial image and the target image. Adding perturbations only at earlier timesteps imposes stricter constraints on keeping the latent trajectory close to the natural manifold, but it also makes the alignment objective harder to optimize. As shown in the following table, under comparable LPIPS, perturbing only the early timesteps (following the update pattern of ACA [2]) leads to poor global similarity on the surrogate model, indicating a more difficult alignment.
>
>
> | Perturbed timesteps (normalized)  | Global Similarity (on suggorate) | ASR (on GPT-5) | LPIPS |
> |--------------------------------------------|:-----------------:|:-----------:|:-----:|
> | only at 0.2                                |       0.70        |    57.2%    | 0.204 |
> | from 0.2 to 0 (original PSI)                             |       0.82        |    78.6%    | 0.192 |
>
> ii) Our PSI balances this trade-off by spreading perturbations across multiple timesteps, instead of applying them to a specific timestep. This avoids imposing a rigid constraint, and helps keep the latent trajectory reasonably close to the natural manifold. Meanwhile, it also provides sufficient flexibility to satisfy the alignment objective.
>
>
> >**Q7: What potential directions exist for defense this diffusion-based attack?**
>
> Thanks for the comments. We discuss this issue from three perspectives: adversarial purification, adversarial training, and adversarial detection.
>
> i) Purification-based defenses, such as DiffPure, can mitigate a broad range of adversarial attacks, including PSI, as shown in Table 2 of the main paper. However, these methods require substantial additional computational overhead and inevitably removes not only adversarial signals but also fine-grained visual details.
>
> ii) Adversarial training shows only limited protection against PSI in our experiments (Table 1 of the main paper). Due to a robustness-accuracy trade-off, adversarial training usually adopt a small perturbation budget [3]. However, this robustness typically does not extend to larger budgets or semantic, diffusion-style shifts like PSI.
>
> iii) Adversarial detection remains an important direction. As shown in Table 4 in Appendix B.4, LVLMs already produce a strong inconsistency signal on PSI even without auxiliary image transformations. This type of detection does not depend on the specific form of perturbation. A promising direction is to investigate whether LVLMs can detect adversarial behavior during normal inference, without additional model calls with specific prompts.
>
>
>
> **Reference**
>
> [1] Mao, Chengzhi, et al. "Understanding zero-shot adversarial robustness for large-scale models." arXiv preprint arXiv:2212.07016 (2022).
>
> [2] Chen, Z., Li, B., Wu, S., Jiang, K., Ding, S., & Zhang, W. (2023). Content-based Unrestricted Adversarial Attack. NeurIPS 2023.
>
> [3] Addepalli, S. et al. Scaling Adversarial Training to Large Perturbation Bounds. ECCV 2022.

---

### Author Response · Authors · 2025-11-20
**Update Manuscript**

We would like to thank all reviewers again for their constructive feedback. We have uploaded a revised version of the paper and highlighted the major changes in blue for clarity. In summary:

1. We have elaborated the Threat Model section in Appendix A.1.

2. We have updated the detailed pseudocode in Appendix A.2.

3. We have theoretically and empirically analyzed the design of PSI in Appendix A.3 and Appendix B.5.

4. We have added an extended ablation study in Appendix B.6.

5. We additionally review representative classic adversarial attacks and adversarial attacks on VLP in Appendix C.

Kind regards, Authors

---

### Author Response · Authors · 2025-12-01
**Author summary comment**

**TL;DR.** We propose PSI, a transferable and stealthy adversarial attack on black-box LVLMs such as GPT-5 and Grok-4. With a clarified threat model, expanded related work, reinforced theoretical and empirical justification, and additional experiments, the revised paper fully addresses the key concerns raised by reviewers.

We summarize the reviewers’ key concerns and our corresponding resolutions as follows:

---

**Reviewer CND6:** Key concern is the trade-off between naturalness and target semantic control in PSI.

**Resolution:** We analyzed this trade-off and showed that distributing perturbations across timesteps improves both global similarity and ASR, clarifying how PSI balances these objectives.

---

**Reviewer orSX:** Concerns remain about coverage of related work and clarity on surrogate–target differences.

**Resolution:** We broadened the related-work review to include classical, unrestricted, and VLP attacks, and clarified differences in threat-model assumptions. We further detailed accessibility and architectural differences between surrogate CLIP models and LVLM systems.

---

**Reviewer EmtG:** Concern focuses on validation of the naturalness term in the joint objective.

**Resolution:** We added PRO-based out-of-distribution analysis and provided empirical justification for incorporating the naturalness term.

---

**Reviewer aN48:** Key concerns are the need for stronger theoretical motivation and more experiments.

**Resolution:** We added theoretical justification for PSI’s perturbation scheme and expanded experiments, including computational-cost analysis.

---

We respectfully point out that although Reviewer orSX gave a score of 2 that is not aligned with the scores of 6 given by the other three reviewers, the concerns raised in that review have been fully clarified during the discussion period and effectively resolved in the rebuttal and revised manuscript.

---

### Meta-Review · Area_Chair_jrAa · 2025-12-29

**Summary:**

This paper studies an important and practical problem: transferable and stealthy adversarial attacks on large vision–language models. The proposed PSI method is carefully designed and shows strong empirical performance on a wide range of open-source and commercial models. The Reviewers raised concerns mainly about the clarity of contributions, the incremental nature of some components, and the need for more analysis and explanation. The authors responded with additional experiments, ablations, and clarifications that substantially address these concerns. Overall, the work is technically sound, the results are convincing, and the paper provides useful insights for the community, so I lean toward acceptance.

**Reviewer Concerns:**

The rebuttal addressed most of the main reviewer concerns. The authors clarified the threat model and attack setting, explained differences from prior VLM attacks, and added new experiments and ablations to support the effectiveness of progressive alignment, co-evolving selection, and the naturalness term. Questions about evaluation discrepancies and robustness were also answered with additional analysis.

Only a small number of concerns remain. Some reviewers may still view the novelty as incremental or wish for stronger theoretical justification, but these points are largely a matter of perspective rather than correctness. Overall, the remaining issues are minor and do not significantly weaken the contribution.

**Reviewer Scores:**

There are four reviewers with initial scores of 6, 2, 6, and 6. After the rebuttal, the three reviewers who gave a score of 6 would likely feel that their main concerns were largely addressed and would keep their scores or become slightly more positive. One reviewer maintains a more critical view, mainly on insufficient discussion of related work and key differences between the source and target models for transferbility; these concerns are well addressed by the authors' rebuttal. Overall, the discussion strengthens the majority positive evaluations, and the remaining concerns appear to be differences in emphasis rather than issues with technical soundness.

---

### Decision · Program_Chairs · 2026-01-26

Accept (Poster)